# From Nanoparticles to Cancer Nanomedicine: Old Problems with New Solutions

**DOI:** 10.3390/nano11071727

**Published:** 2021-06-30

**Authors:** Chi-Ling Chiang, Ming-Huei Cheng, Chih-Hsin Lin

**Affiliations:** 1Comprehensive Cancer Center, Division of Hematology, Ohio State University, Columbus, OH 43202, USA; chiang.127@buckeyemail.osu.edu; 2NSEC Center for Affordable Nanoengineering of Polymeric Biomedical Devices, Ohio State University, Columbus, OH 43202, USA; 3Center of Lymphedema Microsurgery, Department of Plastic and Reconstructive Surgery, Chang Gung Memorial Hospital, College of Medicine, Chang Gung University, Taoyuan 33305, Taiwan; minghueicheng@gmail.com; 4Center for Tissue Engineering, Chang Gung Memorial Hospital, Taoyuan 33305, Taiwan; 5Graduate Institute of Nanomedicine and Medical Engineering, College of Biomedical Engineering, Taipei Medical University, Taipei 11031, Taiwan

**Keywords:** nanoparticles, nanomedicines, nanomaterials, nanotechnology

## Abstract

Anticancer nanomedicines have been studied over 30 years, but fewer than 10 formulations have been approved for clinical therapy today. Despite abundant options of anticancer drugs, it remains challenging to have agents specifically target cancer cells while reducing collateral toxicity to healthy tissue. Nanocompartments that can be selective toward points deeply within malignant tissues are a promising concept, but the heterogeneity of tumor tissue, inefficiency of cargo loading and releasing, and low uniformity of manufacture required from preclinical to commercialization are major obstacles. Technological advances have been made in this field, creating engineered nanomaterials with improved uniformity, flexibility of cargo loading, diversity of surface modification, and less inducible immune responses. This review highlights the developmental process of approved nanomedicines and the opportunities for novel materials that combine insights of tumors and nanotechnology to develop a more effective nanomedicine for cancer patients.

## 1. Introduction

The National Institutes of Health (NIH) defines nanoparticles as structures ranging from 1 to 100 nm in at least one dimension, while current nanoparticles in therapeutic application are acceptable up to hundreds of nm. Considering the tissue junction between capillaries (150–200 μm), nanoscale structures exhibit unique properties to enhance reactive areas as well as across cell or tissue barriers [1]. For pharmacokinetic properties, the optimal size of nanoparticles is around 100 nm in a hydrodynamic diameter. Nanocarrier size also affects in vivo fate, with larger particles (>200 nm) majorly accumulating in the liver and spleen. Smaller nanoparticles are limited to tissue extravasations and renal clearance, whereas larger ones are quickly removed from the bloodstream via the macrophages of the reticuloendothelial system [2]. Of course, in vivo absorption and clearance of nanoparticles depend on additional biomaterial composition as well as surface modification.

Currently, nanoparticles are applied to conventional drugs to improve their efficacy and reduce morbidity for advanced cancer therapies. Antitumor cargos are either capsuled or covalently linked to the nanocarrier. The advantage of covalent links is a precise number of therapeutical molecules for each nanoparticle, while the encapsulation of materials provides more flexibility. Many antitumor drugs are hydrophobic, posing challenges for physiological uptake (Table 1). Hydrophobicity of small molecules leads to easy penetration of cellular membrane, but lower solubility crossing the aqueous phase in circulating or tissue fluids. The low solubility in the aqueous phase usually causes aggregation and local embolism. Conversely, hydrophilic agents, including proteins, peptides, and nucleic acids, are very soluble in body fluids but poorly across the cellular membrane. In the past 10 years, nucleotide-based therapeutics, such as DNA, siRNA, and microRNA, have emerged in preclinical work to regulate certain gene expressions. However, these negative-charged molecules have poor cellular uptake and fast degradation physiologically, limiting their biomedical applications. To solve these problems, various synthetic nanomaterials have been developed. Cationic liposome and polymer are the most common methods to neutralize polyanionic nature of nucleotides in delivery [3,4]. Due to rapid clearance of cationic compartments from circulation, several adjustments were added to these nanocomplexes, finalizing them as neutral nanoparticles or a slight negative charge, which can prolong circulating half-lives [5].

Besides stabilizing anticancer agents, designed nanoparticles can also enhance the delivery efficacy by targeting cancer lesions. This concept led to variable nanoparticle designs fitting physicochemical properties via surface modification for a multitude of biomedical applications. The targeting ability of nanoparticles, either passive or active, is aimed for enhancement of drug concentration within the specific tissue of interest, such as tumors, while limiting toxicity to healthy organs. Passive targeting depends on pathophysiological characteristics of tumor vessels, enabling nanomaterials to accumulate in the microenvironment. In tumor tissue, fast angiogenesis with highly disorganized and loosened vessel structure leads to enlarged gap junctions between endothelial cells, resulting in enhanced permeability and retention (EPR) effect [6]. The EPR effect allows diffusion of molecules less than 400 nm in diameter, which is suitable for nanoscale complex. The other phenomenon generally observed in tumor tissue is the Warburg effect, a local high metabolic and glycolysis rate result in an acidic environment [7]. Designed pH-sensitive biocarrier could be stable at physiological pH = 7.4, but rapidly disassembled and released payload once it reaches an acidic microenvironment. Common design of pH-sensitive nanoparticle is based on polymers with pKa in the range of 6.5–7.2, such as poly(L-histidine) (PHis) and poly(β-amino esters). Bae et al. prepared pH-sensitive micelle with PEG-PHis and PEG-PLA polymers that performs stable at pH 7.0–7.4 and disassembled at a pH of 6.6–7.2 [8]. Furthermore, T. Mizuhara et al. developed a novel pH-responsive alkoxyphenyl acylsulfonamide nanoparticle which become positively charged at pH < 6.5, making pH-controlled uptake and toxicity for tumor selective therapy [9]. The passive targeting fits basic clinical application well but still has limitations. Ubiquitous recognition and off-targeting by passive delivery could cause lower affinity and random binding to the healthy supporting tissue nearby tumors. In addition, the passive strategy faces the obstacle of reaching circulating tumor cells due to the lack of these permeable structure.

Unlike passive targeting, active delivery incorporates other high-affinity molecules to recognize cells directly. Active targeting based on surface receptors on target cells has been widely explored since malignant cells upregulate certain tumor-preferred receptors. For example, transferrin receptor (TfR) and folate receptors (FRs) are physiologically expressed on various normal cells but overexpressed in many cancer types in response to their higher metabolic rate [10,11]. Transferrin and TfR-mediated drug delivery were applied for circulating leukemic cells, but the expression on other normal blood cells might be concerned [12]. Moreover, FR is overexpressed in ovarian, breast, lung, colon, kidney, and brain tumors [13], while folic-acid-conjugated nanoparticle has been reported to have an efficient targeting to the folate receptor of tumor cells in xenografted mice model [14,15]. Due to the different origins of malignancy, the markers of cluster of differentiation (CD) are utilized to reach of cancer cells. B Yu et al. reported a combination of monoclonal antibodies of anti-CD37 and CD19/CD20 as a dual-ligand immunoliposome for efficient targeting of chronic lymphocytic leukemia [16,17]. To advance selectivity, oncofetal antigens provide alternative anchors, since they are produced by tumors and fetal tissues, and a much lower amount by adult tissues. R Mani et al. reported an anti-ROR1 (tyrosine kinase-like orphan receptor 1) immunoliposomal delivery targeting of chronic lymphocytic leukemia and mantle-cell lymphoma B cells but not normal B cells [18,19,20]. Being an ideal targeting antigen or surface molecule, it also requires the homogeneous expression in disease stages or to not be secreted into the blood circulation. In clinical, loss of CD20 antigen expression was observed in various B lymphoid malignancy in recurrent or relapse patients, which would increase the difficulty of continuing targeting them. Furthermore, the ability to stimulate internalization is the other critical index for nanocomplex. The caspase reaction of conjugated complex binding to the surface antigens, triggering target-mediated endocytosis, and releasing from late endosome lead cargo into cytosol [21]. The current potential targets of cancer-relative antigens are shown in the Table 2.

Conjugation is the process to join the recognition molecules with the therapeutical complex, including direct conjugation or indirect method via linker. One of the main challenges in conjugation design is homogeneity of the molecules. Direct conjugation strategies require nanoparticles with functional groups such as amine, aldehyde, or active hydrogen group on the surface. For example, the immunoglobin or targeting ligand can react with DSPE-PEG-amine phospholipid and then be inserted to the surface of liposome [22]. The vicinal diol on the carbohydrates residues of immunoglobin is a perfect target for aldehyded nanoparticle and carriers to form a hydrazide bond [23]. The hydrazone ligation, a novel conjugation method developed since the 2000s, improves the conjugation with high purity and low side products. Because the natural biological molecules contain the rare functional groups forming the hydrazone bond, the ligation happens very selectively. By using a hydrazone ligation, Dawson et al. synthesized viral nanoparticles and conjugated with VEGFR-1 ligand (F56f peptide) on benzaldehyde cowpea mosaic virus nanoparticle for tumor targeting and imaging [24]. Moreover, considering orientations of ligands or antibodies; thus, conjugation via linker chemistry is better than direct conjugation for targeting molecules to nanoparticle. The conventional linker relies on the reaction between amine-modified nanoparticles and sulfhydryl-containing biomolecules, which may result in a heterogeneous mixture in the drug-carrier-antibody ratio and drug load distribution/location [25]. Recently, several studies have been made in the field of site-specific conjugation with linker to improve the synthesis of the homogeneous conjugation, such as engineered reactive cysteine residues, unnatural amino acids, aldehyde tags, and enzymatic transglutaminase- and glycotransferase-based approaches [23,26,27,28].

The internalization of nanoparticle is regulated by endocytosis pathways based on its dimension. The nanoparticle with a larger size around 150 nm was up taken by clathrin-mediated pathway, while smaller nanoparticles of about 50 nm by clathrin-independent endocytosis [29,30]. In both of clathrin-mediated and -independent pathways, the intracellular nanoparticle is fused with early endosomes and late endosomes (pH~5) for the membrane recycling. After the late endosome stage, the cargo needs to be released by either acidic pH or enzymes and to have the effect on target cells. Although the active targeting strategy can increase the uptake which facilitates the nanocarriers internalization, the challenges still exist for designing the best agents and techniques to facilitate the endosomal escape of the cargo within the cell [31]. Several endosomal escape-enhancing strategies need to be considered in the context of the interaction of nanoparticle with the special endosomal environment. These include but are not limited to usage of cationic polymers, pH sensitive polymers, and calcium phosphate for escaping [32]. The novel material of cyclic heptapeptide cyclo(FΦRRRRQ) (cFΦR4) was recently reported to be efficiently internalized and escaped from early endosome, providing a useful transporter for intracellular cargo delivery in mammalian cells [33]. A novel carrier of nanodiamond was reported to enter the cells via endocytosis and quick escape from endosome by rupturing its membrane to accomplish a successful cytosolic delivery [34].

Overall, development of nanomedicine from past decades is a proof of concept to selectively increase the concentration of anticancer agents in tumor malignancy but minimize the side effect from healthy tissues (Figure 1). In the following section, we discuss several nanomaterials currently used in biomedical by their classification: lipocomplex, polymeric nanoparticle, carbon nanotube, DNA origami, and exosome-derived vehicle. We highlight their material properties, biomedical application, preclinical or clinical usage, and recent challenges (Table 3).

## 2. Current Materials in Nanomedicine

### 2.1. Lipocomplex

Liposomal nanocomplex is the first delivery tool since the first discovery in the 1960s by A.D Bangham’s group. Liposome formulation ranges from 50 to 200 nm with spherical vesicles composed of phospholipids, and steroids form bilayers in aqueous media can benefit as biocarriers [35,36]. The properties of liposome were simply applied to increase the solubility of hydrophobic molecules and accelerate physiological metabolism in the beginning. For example, plenty of liposome formulations tried to fit numerous biochemical agents and provide less toxic than the free form. Liposomes were used to deliver lysophosphatidic acids and its analog which regulate normal or malignant blood cell differentiation and proliferation [37,38]. However, the liposomal formulations in this period faced a severe problem of short pharmacokinetic half-life, until the “stealth liposomes” was designed the 1990s. The second generation of liposome introduced the surface polyethylene glycol (PEG) coating, which highly improved stability and longer circulation time by alleviating the uptake of macrophages [39,40]. The PEGylation, constructed with a hydrophilic film on surface, can protect the liposome from clearance of reticuloendothelial system, making liposomal delivery clinical practical.

Liposomal structure delicately contains both hydrophilic lipid bilayer and hydrophilic inward, providing the flexibility as a perfect vehicle. Hydrophobic compounds insert within liposomal lipid bilayer, while hydrophilic substances are encapsulated in the internal water phase [41]. The lipid film formation is a common method that incorporates the hydrophilic cargo with a thin lipid film to self-organize to make bilayers. The advantage of this method is its simplicity, but only a low percentage of hydrophilic cargo can be encapsulated [42]. The freezing-and-thawing technique, which uses a sequence of freezing and thawing to form transient holes by ice crystals, provides better drug penetration, while increasing the final volume of lipocomplex [43,44]. The ethanol injection is the other alternative method by using a rapid injection of drugs to make unilamellar liposomes encapsulated with drugs [45]. The encapsulation efficiency of ethanol injection is still low, but combination with positive charged carriers, such as polyethylenimine, highly improves the encapsulation rate [46,47]. The reverse-phase evaporation method, by mixing the cargo and lipocomplex within emulsion and evaporating solvent after gel formation, yields the highest efficiency by means of passive loading (~50%). However, the remaining organic solvent after evaporation limits the clinical use [48,49]. A modified reverse-phase evaporation method was demonstrated by Handa et al. to high encapsulation efficiency reaching 80% [50]. Due to the limitation of loading rate of current methods, the free drug outside liposome must be removed by dialysis or molecular exclusion chromatography. Although liposome is widely used in clinical application, there is a challenge in maintaining size uniformity during the manufacturing process. Generally, the uniformity and stability of liposome are majorly decided by the formulation and characteristics of lipocomplex. Using lipid extrusion, which forces the liposomal suspension through a series of polycarbonate filter, typically 0.1–0.8 μm membranes under high pressure (>500 psi), makes the particles with a diameter near the pore size of the filter applied. However, the rigidness and elasticity of the liposome passing the nanopore of the filter are determined by the composition of lipocomplex.

Several lipid complexes have been approved for clinical treatment after fifty years studying of lipocomplex (Table 4). In 1995, the first liposomal pharmaceutical product, Doxil (Ben Venue Laboratories, Inc.), was approved by the US Food and Drug Administration (FDA) for the treatment of chemotherapy-refractory acquired immune deficiency syndrome (AIDS)-related Kaposi’s sarcoma [51]. The main purpose of Doxil in clinical is to avoid the severe cytotoxicity for cardio tissue, although the clearance of liver and kidney may reduce the dose in circulation, causing lower efficiency in lesions. Currently, more than 12 liposome-based drugs are approved for clinical use and more are in stages of clinical trials [52,53,54]. Generally, the liposomal formulation drugs are approved for intravenous treatment. The availability of doxorubicin hydrochloride liposome injection was approved to supply the U.S. patients for treatment of ovarian cancer and Kaposi sarcoma of AIDS in 2013. Recently, CPX-351, a liposomal formulation of cytarabine and daunorubicin, receives its fast track designation as a secondary AML drug in elderly patients [55,56]. Moreover, lipocomplex provides more flexibility for drug combination for clinical use. Immune therapy with monoclonal antibody of Bevacizumab (Avastin, Genentech, Inc.) combing with paclitaxel, pegylated liposomal doxorubicin, or topotecan was currently approved for the treatment of patients with platinum-resistant, recurrent epithelial ovarian, fallopian tube, or primary peritoneal cancer [57]. However, further improvement is still required to overcome the limitations of liposomal therapy facing today in terms of long-term stability, efficient drug loading, and active targeting.

The gap between the small laboratories’ models to large industrial units available for clinical suppliers still need to be filled. To avoid the oxidation and chemical reaction during the long-term storage and delivery, freeze-drying has been a standard practice for liposomal industry. However, the removal of solvent from liposomal solution by lyophilization and reconstitution later may cause the leakage of encapsulated materials, posing a challenge for wide application of liposome as drug carrier. Certain cryoprotectants are used during the process, such as trehalose and sucrose, to retain the liposome and cargo as much as their original ratio and substances. Otherwise, the loading efficiency of liposomal industry depends on the solubility of the cargos in either liquid or lipid phase. Generally, passive encapsulation is limited by the tiny trapped volume within the liposomes, which is lower than 30%. To promote efficiency, cargos sometimes are modified with protonizable amine to actively cross pH gradient generated between liposomes bilayer [58]. For example, the current active loading method for liposomal doxorubicin which can reach a high efficiency (~92%) of the total drug was encapsulated, largely decreasing the waste of unencapsulated drug [59].

### 2.2. Polymeric and Dendrimer Nanoparticles

Polymeric nanoparticles (PNPs) are structures with a diameter ranging from 10 to 100 nm, which was made from synthetic polymers (e.g., polycaprolactone and polyacrylate) or natural polymers (e.g., albumin, chitosan, and gelatin) [60,61]. Clinical application of PNPs has reduced ionic surface to avoid the immunological response, while the immobilization of drug within PNPs can increase the drug stability as well. For example, docetaxel-loaded polymeric micelle (diameters < 30 nm) can reach poorly permeable pancreatic tumors in vivo [62]. The enhanced stability of the immobilized drug is attributed to the interaction with the polymer carriers to avoid the degradation. Once the complex reaches the target tissues, release mechanism would be triggered by tumor microenvironment. Several unique properties of tumor microenvironment have been used for cargo releasing, such as acidic, hyperthermia, and special enzymes secretion in the local environment. The pH-sensitive polymers are relative stable at a physiologic pH of 7.4 but can be rapidly destructured and can release active drugs in acidic tumor tissues. For example, poly(lactide-co-glycolide) (PLGA) polymer performed as 2–4-fold doxorubicin release in tumor-bearing tissue than circulation at pH 7.4 [63]. Moreover, thermosensitive polymeric, such as poly (N-isopropylacrylamide- co-acrylamide-co-allylamine) (PNIPAM-AAm-AA), could be a potential anticancer drug nanocarrier. Under the hyperthermia of tumor region, the hairy structure of PNIPAM-AAm-AA polymer would shrink, while the enclosed doxorubicin releases rapidly [64,65]. Additionally, cancer cell secret unique enzymes such as matrix metalloproteinases (MMPs) to help themselves in migration and metastasis. An MMP-activatable peptide-conjugated polymer and drug has demonstrated an efficiently and specifically cleaving by MMP secreted by cancer cells in vitro study [66,67]. The release mechanism is suitable for polymer with relatively low molecular weight, while the departure from higher-weight polymer is slower. However, for synthetic polymer, challenges in reproducibly maintaining the same size and molecular weight from each batch led to the development of dendrimers. 

Dendrimer is a unique structure of polymer, which was first synthesized by Vogtle group in 1978, with branched 3D structure that provided a high degree of functional surface [68]. This multifunctional property provides the dendrimers more loading space for cargos and interaction with target cells. The cytotoxicity of dendrimer carrier depends on its surface area and the arms of dendrimer, while exchanging the amine groups into hydroxyl group may result in lower levels of cytotoxicity in vivo. The drug could be loaded into the internal structure of dendrimers or covalently linked to dendrimers molecule. Compared to the linear polymers with stochastic structures, dendrimers offer a well-defined size and structure, performing a more precise polyvalence and molecular weight. The polyvalence defines the exact number of active groups on a single dendrimer. By controlling the number of covalent bonds within a single molecule, the quantity of drug loading could be adjusted. Noncovalent encapsulation is an alternative method only when payload is labile or poorly soluble. Poly(amido amide) (PAMAM), a very common dendrimer widely used in biomedical applications, is easily to have molecular conjugation through its branches of amine terminals [69]. Thioaptamer (TA)-modified PAMAM is developed to target CD44^+^ (TA receptor positive) breast cancer in vitro and in vivo by using ligand-receptor affinity [70]. Moreover, introducing a folic-acid conjugation has been reported to improve the delivery of PAMAM dendrimers loaded with 2-methoxyestradiol to target *KB* carcinoma cells overexpressing high-affinity folic acid receptors [71]. Interestingly, additive PEGylated coating may decrease the toxicity of dendrimer. Liu et al. demonstrated a dual-functionalized dendrimers with PEGylation and thiolation improving blood compatibility [72]. This process extends the lifetime of dendrimer in blood circulation to avoid unnecessary accumulation in normal organs, such as kidneys and liver.

Unlike liposome, which has had clinical application for over two decades, the medical application of synthetic polymer in drug delivery is just emerging. Here, we highlight paclitaxel (PTX) as a great example for combination of polymer and anticancer agents in pharmacotherapeutic industry. PTX is known as a powerful anticancer agent for various cancers, but treated patients usually face serious neutropenia and sensory neuropathy. These adverse reactions are attributed to the mixture of Cremophor EL and ethanol, which is the special solvent in clinical usage for the very hydrophobic agent, PTX. This suggests the effort to devote to alternative formulation or nanocarrier for PTX. However, PTX, as a very hydrophobic agent, is supposed to be embedded within the lipid bilayers of liposomes, but its bulky and asymmetric structure makes loading difficulty. Therefore, plenty of polymer-based nanocarriers are designed to solve the issue by burying PTX in its hydrophobic core and deliver within a whole soluble complex. Abraxane (ABI-007), the co-condensate of natural polymer albumin complex and paclitaxel, was a very successful nanomedicine to solve the problem as approved by the FDA in 2013 [73]. Abraxane demonstrated significantly better tumor killing and longer times of tumor progression in patients of metastatic breast cancer, but the utility needs to be improved in the binding ability between drug and albumin polymers. Other designs of PTX-polymer nanomedicine are ongoing in clinical trials: CT-2103, polyglutamate-conjugated PTX (ovarian cancer, NCT00108745) and Genexol-PM micelle (pancreatic cancer, NCT00111904). Genexol-PM, an improved polymeric micelle formulation to carry paclitaxel, has been approved by the FDA in clinical treatment for pancreas, breast, and small cell lung cancer, while serial studies are currently underway with phase III and IV [74,75]. Moreover, other dendrimer nanocarriers, such as OP-101 (dendrimer N-Acetyl-Cysteine, NCT03500627) and ImDendrim ([188Re] Rhenium associated dendrimer), are just entering the very beginning of clinical trial for its safety in phase I.

### 2.3. Carbon Nanomaterials

Carbon nanotubes (CNT), widely used as nanocarriers, are characterized by the unique structure with the rolling of a single (SWCNTs—single-walled carbon nanotubes) or multi (MWCNTs—multiwalled carbon nanotubes) sheet of graphite with an enormous surface area and an excellent electronic and thermal conductivity [76]. The compatibility of nanotube could improve biomedical reagent delivery with advanced chemical modification on its surface. SWCNT has a defined wall, whereas MWCNT mostly has structural defects which result in a less stable nanostructure [77]. SWCNTs is a one-dimensional nanomaterial composed of a single graphene layer of cylinder shape in a diameter of 1–2 nm and a length ranging from 50 nm to hundreds of μms. SWCNTs exhibit higher accumulation in tumor tissues physiologically, and their needle-like shape facilitates transmembrane penetration and internalization of therapeutic cargos. Moreover, a high surface area enhances ability to encapsulate and load cargo onto their surface or within their interior core via both covalent and noncovalent linkage. As drug carriers, there remain advantages and disadvantages of SWCNT relative to MWCNT. The stronger structure of SWCNT might be suitable for quality control of delivery, while the low stability of MWCNT makes it easier for further modification. Al Faraj et al. have recently demonstrated enhancement of delivery of doxorubicin by antibody-conjugated magnetic SWCNTs, which can also perform as a noninvasive imaging biomarker [78,79]. A. Pistone et al. have currently demonstrated hydroxyapatite-magnetite with MWCNT as a biocompatible magnetic drug delivery system in bone tissue engineering [80].

For nucleotide delivery, several studies evaluated the ability of CNTs which are covalently or electrostatically linked to nucleotides as transfection agents to deliver gene materials [81,82]. However, direct covalent conjugation with CNT may lead to poor intracellular release of nucleotides, thus limiting the biological function. The alternative option is to use functional cationic CNTs, linking polyethyleimine or pyridinium with CNT, to carry negatively charged nucleotides. Siu et al. have recently reported a noncovalently functionalized SWCNT-polyethyleimine for topical siRNA in vivo delivery into melanoma [83]. Moreover, to improve the ability of penetration, the carbon nanostructure could be modified into horn-shaped sheath aggregation with similarity of graphene monolayer. An oxidized single-wall carbon nanohorns, entrapped cisplatin for anticancer, was reported with a property of slowly releasing in aqueous environments, providing an effective suppression of human lung cancer cells [84]. The safety report indicated that CNTs are low in toxicity under the dosage (60 mg/kg) in mice, but it still lacks the further pharmakinetic studies in human [85]. The possible toxicity of long-term exposure to carbon nanomaterials are free radical formation, generation of radical oxygen species, increased inflammatory responses, and granuloma formation. The application for using carbon nanomaterials in clinical use requires more investigation for long-term treatment.

### 2.4. Nucleotide-Based Origami

DNA origami technique to build up uniform nanostructure was first named and introduced by PWK Rothemund in 2006. The method is to establish a scaffold which folds DNA into a desired shape using hundreds of short complement staple strands [86,87]. In the 2000s, DNA origami was widely investigated as candidates to serve as the next-generation drug-delivery vehicle [88]. Compared to other nanoscale methods for drug delivery, such as lipocomplex and inorganic nanoparticles, nucleotide-based origami performs several advantages: (i) uniformity of size, shape, and charge for each particle with self-assembled nanostructures; and (ii) precise control of the cargo loading on the scaffold by specific oligos or functional groups. The small DNA nanocarrier could serve as an effective delivery tool for anticancer drugs, RNA interference reagents, oligo-DNA, and antigen molecules, either in vitro or in vivo. Jiang et al. first showed a high level of doxorubicin loaded in DNA origami, and the complex exhibited prominent cytotoxicity in human breast cancer cells (MCF 7) and doxorubicin-resistant cancer cells [89]. Then, Zhang et al. further demonstrated the in vivo effect of a doxorubicin-containing DNA origami exhibited antitumor efficacy without observable systemic toxicity in nude mice bearing breast tumors [90]. The stability of DNA origami is necessary for applying drug delivery. Within the physiological condition, the scaffold of DNA origami was stable after 12 h incubation and can be slowly degraded in living cells for 72 h treatment [91,92]. Moreover, the DNA origami could serve as an intracellular pH-sensitive nanocarriers loaded with small interfering RNA which are specifically released in human cancer cells [93]. The potential of DNA origami constructs was tested, showing programmable, noncytotoxicity, but recognition by Toll-like receptor 9 (TLR9), which is a receptor for innate immune system localized in the endosome [94]. However, the recognition of double-stranded DNA of origami structure by TLR9 in the endosome would stimulate a strong immune response against virus, posing a challenge for its in vivo application. The immune compatibility would be critical for the development of next generation of DNA origami.

Deposit a great strength under physiological pH condition, the structure of DNA origami has relative limited thermal and chemical stability. A considerable serum-induced degradation of 3D DNA origami structure was observed, which includes rapid collapse and then reaches a slow degradation phases [95]. To prevent degradation and to stabilize the structure, a couple modifications could be applied to the nucleotide scaffold. For instance, locked nucleic acids are designed with an extra methylene bridge between the 2′-oxygen and 4′-carbon of ribose to have a stable A-form helix, which can significantly increase both thermal stability and nuclease resistance. Other chemical modifications, including 2′-O-methyl, 2′-amino, 2′-fluoro, and phosphorothioate substitutions, can also increase the stability of the nucleotide backbone. Moreover, the nucleobases could be modified, such as the hydrogen-bonded base pair of Z or P (6-amino-5- nitro-3- (1′-β-D-2′-deoxyribofuranosyl)-2(1H)-pyridone and amino-8- (1′-β-D-2′- deoxyribofuranosyl)- imidazo[1,2-a]-1,3,5- triazin-4(8H)-one, respectively, which increases hybridization specificity. These materials can be used to site-specifically decorate DNA scaffolds with their canonical Watson–Crick base pairs to increase the stability of origami structure. Recently, Y Zhao’s group has a very exciting design of intravenously injected DNA nanorobots (90 nm × 60 nm × 2 nm) that deliver thrombin specifically to tumor-associated blood vessels and induce intravascular thrombosis, resulting in tumor necrosis and inhibition of tumor growth [96]. The DNA nanorobot not only performed a great targeting to tumor vessel via nucleolin-targeting aptamer but also no significant impact on cytokine levels (IL-6, IP-10, TNF-α, and IFN-α) in preclinical annual studies.

### 2.5. Exosome-Derived Vehicle

Exosomes are cell-derived vesicles, ranging from 30 to 150 nm, that are present in many and perhaps all biological fluids for cellular communication. Exosomes were first described by Trams et al. and later substantiated by Johnstone et al., who observed intracellular interaction with small particles [97,98]. The main function of exosome was suggested as a route of cellular communication, which allows cells to exchange biomaterials, such as RNA, proteins, and lipid components. Since it is composed of partial cellular membrane, implying properties of high compatibility, low toxicity, and limited immunostimulation, exosomes are now regarded as a potential carrier of cargos to be delivered to the secondary cell. The lipid composition of exosome shares certain similarity to parental plasma membrane, but a different lipid raft composition with increase in sphingomyelin, phosphatidylserine, phosphatidylglycerol, lyso-phosphatidylethanolamine, and lyso-phosphatidylchoxline [99]. Due to the negatively charged phospholipid membrane, exosomes show negative zeta potential from –10 to −70 mV in physiologic pH [99,100]. Moreover, the molecular composition is highly dependent on the parental cell type, while the CD63, CD9, CD81, and CD31 are general surface biomarkers [100,101,102,103,104].

The utility of exosomes as a potential drug delivery vehicle relies on a stable and quantified cell source for exosomal production. For example, human mesenchymal stem cell is a promising exosomal producer, which is not only an easily accessible cell type but is also highly proliferative [105]. Moreover, the MSC-produced exosome is attractive due to its safety in clinical application. More than 500 MSC relevant clinical trials have been tested since 2010 [106]. The dendritic cell-derived exosomes caught much attention as immunotherapeutic anticancer agents based on their functional MHC–peptide complexes, which facilitate immune cell-dependent tumor suppression [107,108]. Moreover, Alvarez-Erviti et al. produced a self-dendritic cell-derived exosome conjugated with neuron-specific RVG peptide to pass the blood–brain barrier, proposing an efficient drug delivery to the tumor in central neuronal system [109,110]. The quality and quantity of exosomal production could be easily controlled, while the producer cells are immortalized to create permanent cell lines. Although the methods provided a rapid and productive way to provide exosome, the chunk of DNA, RNA, and protein from producing cells would remain in the artificial exosome, which may also affect target cells in the clinics. How to remove the original cargo within exosome is still a challenge that needs to be overcome in this field. Moreover, the loading and excretion mechanism of exosome is ambiguous [110,111]. Currently, passive loading methods, such as electroporation and positive-charged carrier, were frequently used [112,113]. The encapsulation of anticancer drugs within an exosome could also have difficulty since the cytotoxicity could affect the parental cells directly. Therefore, SC Jang et al. utilized an exosome-mimetic nanovesicle to encapsulate doxorubicin in vitro rather than made through excretion from cell source [114]. An efficient method for exosomal loading is essential for the developing exosome as a future practical nanocarrier in biomedical application [115].

Most clinical usages of exosomes in cancer medical remain in basic observation, early diagnosis, or prognosis after treatment. The first exosome-based cancer therapy is a vaccination therapy, which started in 2010, with tumor antigen-loaded dendritic cell-derived exosomes against advanced unresectable NSCLC patients (phase II, NCT01159288). Furthermore, the other trials recruit plant exosomes to deliver natural agents or nutrition to head and neck or colon cancer patients (NCT01668849 and NCT01294072). Overall, the field of using exosomes as a nanocarrier is still very new.

## 3. Summary of Clinical Trial

Various nanoparticles have been approved for clinical use, either by the FDA in the United States or the European Medicines Agency in the European Union. These nanocarriers are all liposomal base with encapsulating an anticancer drug, except Abraxane with albumin. It takes more than three decades for Doxil, the first FDA-approved cancer nanomedicine in 1995, since the first liposome was described in 1965 [35,36,116]. After that, other liposomal formulations were approved by the FDA and albumin-bound nanoparticle for cancer treatments [73]. However, these formulations are all passively targeted with less active delivery or targeting ability. Even then, the advantages, reduced toxicity, selectively acculturation at tumor sites with limit off-target via EPR effect, and increased efficacy still proved these nanoparticles successfully over their free drug molecules in clinical trial. Given the successes of nanomedicine in the clinic, significant efforts continue to expand the approved nanomedicines to new therapy as well as developing new formulations. For example, Doxil, early proven decades ago, is now included in over 200 clinical studies for additional combination with variance immunotherapy. Newly approved nanomedicines, such as Marqibo and Onivyde, are also participating in plenty clinical trials now, seeking for additional cancer types, combination therapies, or upgrading from a secondary to a first-line therapy. In Table 5, we included the summary of several ongoing clinical trials of nanomedicine.

Advances in nanomedicine and material science could be a big game changer to overcome the existing limitations of microRNA and gene therapy. RNA interference and microRNA were discovered in 1993 by Lee and colleagues with specific regulation of gene expression in test tube, while keeping stagnation as the only prognostic biomarker in clinical for decades [117]. Short-circulation half-life limited cellular uptake and off-target effects on nondesirable tissues pose a challenge for improving miR as direct therapeutic molecules. Therefore, an alternative delivery method is required. Packaging nucleotide cargos into a nanocarrier for therapeutic applications of cancer realm is beginning to enter the clinic trial. MRX34 (Mirna Therapeutics), a potential first-in-class miRNA mimic therapy for cancer, is the first liposomal formulation of the naturally occurring tumor suppressor miR-34a [118]. MRX34 entered a multicenter phase I trial of solid tumors in 2013 with significant tumor suppression but was terminated due to certain immune-related serious events [119]. Despite the attempts to encapsulate the miRs in a lipid carrier to enhance the sustainable delivery and accumulation in tumor microenvironment, the severe immune reactions were discouraging [120]. These experiences reiterated the need for targeted delivery of miR formulations with emphasis on selectivity and specificity to avoid possible immune-related toxicities. MesomiR-1, a miR-16-based microRNA mimic encapsulated in EnGeneIC delivery vehicle (EDV), is entering phase I for mesothelioma and non-small cell lung cancer in 2015 [121]. The miR-16 family, known as a tumor suppressor in multiple cancer types, is applied with EDV for cancer therapy [122]. As an alternative delivery to conventional liposomal, EDV is derived by EnGeneIC Ltd. (Sydney, Australia) as nonviable minicells (~400 nm in diameter) by de-repressing polar sites of cell division in bacteria [123]. Once payload is loaded, EDVs are further coated with bispecific antibody, which has one arm with an anti-EGFR antibody (ABX-EGF) and the other with anti-lipopolysaccharide (LPS) antibody (1H10) for anchoring to the surface of minicell [124]. Although there is concern of inducible pyrogenic reactions by bacterial endotoxin product (LPS), the preliminary safety profile encourages TargomiRs for additional studies in combination with other first-line therapies for patients of malignant pleural mesothelioma [125,126]. With various nanoparticle-based gene therapies are performing in clinical trials with shutdown or suppression of certain genes, advanced gene-editing therapies are close behind these systems for further preclinical studies.

The interest of encapsulating and delivering small compounds in clinical trials via nanoparticles is the most abundant. Although most of these studies are following liposomal conception, with similar designs, although modified from approved liposome; still, certain ongoing trials try to introduce novel schemes in the clinic. The direction of these new investigations for nanomedicine fall into three major categories: (a) bring and encapsulate more options of anticancer agents into nanomedicine system for various cancers; (b) targeting and directing nanomedicine toward cancer lesions and releasing payload locally; and (c) combination therapy, encapsulating a well-defined and synergistic ratio of multiple anticancer drugs within one nanocarrier. For example, the great success of abraxane (albumin-particle bound paclitaxel) in cancer medicine encourages numerous other different paclitaxel-based nanomedicine waiting for clinical trials. Interestingly, both nanoparticles and the immunotherapeutic antibody are approved for clinical usage, but a system to combine the two technologies is just recently entering clinical trial, such as the nanocarriers targeting for EGFR (Anti-EGFR-IL-dox), TfR (MBP-426), or PSMA (BIND-014). Another alternation for selective delivery is directing nanomedicine toward cancer lesion by additional stimuli-responsive functions. ThermoDox is designed as a heat-sensitive liposome, which releases cargos upon exposure over 42 °C. This control can be precisely achieved via adding microwave hypothermia or ultrasound locally to the tumor lesions of breast and liver cancer. The other benefit of nanocarrier technology is the potential of combinational therapy. Combinational therapy for multiple free drugs typically faces the challenge of two identical drugs with very distinct pharmacokinetic and interaction properties, while it is practical by delivering precise molar ratios of them to the tumor site synchronically and systematically via one nanocarrier compartment. VYEXOS (CPX-351) is the first combinational therapy by encapsulating a synergistic ratio of cytarabine and daunorubicin (5:1) and now in phase III of clinical results. In August 2017, the FDA approved Vyxeos for first treatment for two types of poor-prognosis AML: newly diagnosed therapy-related AML (t-AML) or AML with myelodysplasia-related changes (AML-MRC). 

## 4. Conclusions

Nanocarriers are designed to improve the pharmacological and therapeutic properties from traditional free drugs. With growing knowledge of tumor heterogeneity and identified biomarkers, new nanomedicines are optimized with efficiency and selection to tumor lesions. From briefly prolonging circulation time to leading anticancer drugs toward lesions, the control of releasing would be the next step. Patients would benefit from the reduction of dosage index as concentrating therapeutic reagents pharmacologically to local tumor tissue and avoiding the universal side effect.

Increasing the need for a new strategy of disease treatment achieves the coordination of diagnosis and therapy by using advanced nanomaterials. The new direction of nanotechnology attempts to integrate therapeutics and diagnostics into a single nanomaterial, referred to as theranostics. The concept of theranostics provide the major applications in clinics which can improve targeted delivery, achieve gene delivery, and have the disease monitoring with the imaging platform by well-engineered nanoparticles. Currently, the pharmaceutical company Cristal Therapeutics is participating in a phase I clinical trial of CriPec^®^ docetaxel combined with the imaging agent Zirconium-89 for PET imaging [127]. The platform evaluates the biodistribution and accumulation of the nanomedicine in solid tumors, leading to a better targeted therapy and follow-up prognosis. In the other clinical trial, Nanobiotix performs phase I/II trials for NBTXR3 comprising hafnium oxide nanoparticles as a radio-enhancer to kill tumor burden by locally additional radiation [128]. This also provides a new perspective to coordinate the imaging and radiology by advanced nanotechnology. Despite considerable development in this direction, nanomedicine of theranostics still faces challenges. The major challenge to successfully translate theranostic nanomedicine into routine clinics is the nano–bio interaction. The therapeutic nanoparticles generally have a larger window of treatment in patients which requires low tolerance of nano–bio interaction, while the diagnostic nanomaterials could be one-dose and real-time imaging every couple months. The cooperation of these different fields of nanomedicine requires further effort on developing innovative nanomaterials to achieve the goal.

Overall, most approved nanomedicines are those developed early and classic antineoplastic, meaning plenty of room for improvement. The next generation of nanomedicines will incorporate more diversity of new small-molecular compounds (pathway inhibitors, such as Rapamycin, a selective mTOR inhibitor) or gene therapeutic agents (siRNA, mRNA and gene editing). This flourishing field of nanoparticle delivery is expected to expand the versatility and potency of nanocarrier for cancer therapeutics. Given recent technical and material advancements in the past decades, smart and precise nanoparticles as drug carriers will revolutionize cancer therapy, not only significantly extending the patient’s lifespan but improving their quality of life. This review has explored the importance of the convergence of nanotechnology and tumor biology from the history of nanoparticles to clinical translation nanomedicine in future. We expect nanomedicines will improve the paradigm of cancer treatment in near future.

## Figures and Tables

**Figure 1 nanomaterials-11-01727-f001:**
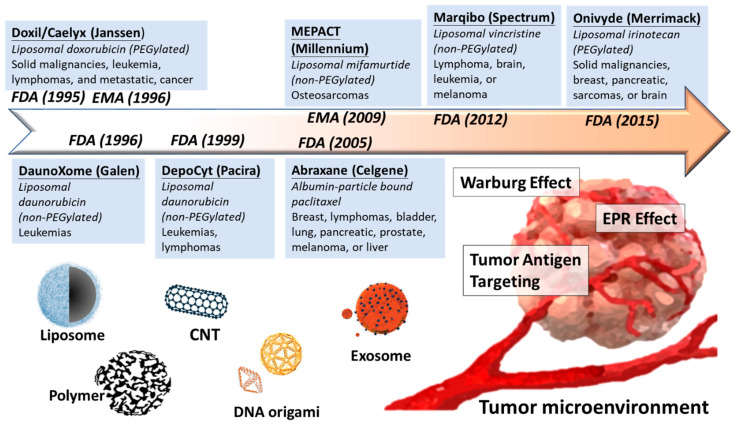
Development for nanomedicine reaching for tumor microenvironment. In past decades, plenty of nanocarriers are moving from preclinical bench work into clinical trial and finally approved for cancer therapy. The driving force of nanomedicine toward tumor microenvironment could be passive or active. Passive delivery relies on loose tumor vessels (EPR effect) and low pH (Warburg effect), while the active delivery can directly recognize tumor antigens by conjugating high-affinity molecules. Various novel and advanced materials of nanocarriers are designed for drug delivery, including liposome, polymer, CNT, DNA origami, and exosome.

**Table 1 nanomaterials-11-01727-t001:** Hydrophobic and hydrophilic anticancer drugs in clinical use.

Drug	Solubility (in Water; 25 °C)	Clinical Use
	*Hydrophobic*	
**Docetaxel**	insoluble (<0.3 μg/mL)	Breast, prostate, non-small cell lung cancer, carcinoma, and adenocarcinoma
**Paclitaxel**	insoluble (<0.3 μg/mL)	AIDS-related Kaposi sarcoma, breast, ovarian, and non-small cell lung cancer
**Alitretinoin**	0.6 μg/mL	Acute promyelocytic leukemia, and AIDS-related Kaposi sarcoma
**Etoposide**	0.03 mg/mL	Small cell lung and testicular cancer
**Cisplatin**	2.5 mg/ml	Testicular, ovarian, breast, glioblastoma, non-small cell lung cancer, malignant mesothelioma, and lymphoma
**Methotrexate**	2.6 mg/mL	ALL, breast, and lung, head and neck cancer, non-Hodgkin lymphoma, and osteosarcoma
**Fludarabine**	3.53 mg/mL	CLL
**Doxorubicin**	10 mg/mL	ALL, AML, neuroblastoma, soft tissue and bone sarcomas, breast, ovary, urinary bladder, thyroid, gastric, thyroid, gastric cancer, Hodgkin’s disease
**Irinotecan HCL**	25 mg/mL	Colon, and rectal cancer
**Cyclophosphamide**	15.1 mg/mL	ALL, AML, CLL, CML, breast cancer, Hodgkin lymphoma, multiple myeloma, and neuroblastoma
**Gemcitabine**	51.3 mg/mL	Pancreatic, breast, ovarian, and non-small cell lung cancer
	*Hydrophilic*	

**Table 2 nanomaterials-11-01727-t002:** Type of cancer antigens for targeting.

Antigen Types	Cancer Antigen	Cancer Type
**Cluster of differentiation**	CD20, CD19, and CD37	Chronic lymphocytic leukemia, and non-Hodgkin lymphoma
	CD33, and CD123	Acute myeloid leukemia
	CD52	Chronic lymphocytic leukemia
	CD30	Hodgkin lymphoma
**Surface glycoprotein**	Epithelial cell adhesion molecule (EpCAM)	Breast, lung, colon, and ovarian cancer
	Carcinoembryonic antigen (CEA)	Breast, lung, gastric, pancreatic, bladder, cervical, and hepatic cancers, lymphoma, melanoma
	Glycoprotein A33	Colon cancer
**Growth factor and receptor**	Epidermal growth factor receptor (EGFR)	Glioma, breast cancer
	Human epidermal growth factor receptor 2 (HER2)	Breast, lung, colon, ovarian, and prostate cancer
	Receptor tyrosine kinase-like orphan receptor 1 (ROR1)	Chronic lymphocytic leukemia, breast, colon, and bladder cancer
	Insulin-like growth factor 1 receptor(IGF1R)	Lung, breast, head and neck, and glioma cancer
	Folate receptor (FR)	Breast, ovarian, and colon cancer

**Table 3 nanomaterials-11-01727-t003:** Properties of different nanomaterials.

	Size Range	Toxicity	Immunogenicity	Cargo Loading	Manufacture
**Lipocomplex**	Wild	Low	Low	Easy	Well
**Polymer**	Precise	Low	Low	Easy	Well
**Carbon Nanostructure**	Precise	High	Medium	Easy (Hydrophobic agents)	Well
**Nucleotide-based Origami**	Precise	Medium	Medium (e.g., Toll-like receptors)	Easy (Hydrophilic agents)	Well
**Exosomeal Vehicle**	Wild	Very low	Very low	Hard	High cost

**Table 4 nanomaterials-11-01727-t004:** Nanotherapeutics approved for oncological therapy.

Name	Particle Base	Anticancer Drug	Cancer Type	Approval
*Liposome-based*				
**Doxil/Caelyx** (Janssen)	PEGylated liposome	Doxorubicin	Ovarian, breast cancer, leukemia	FDA, 1995
**DaunoXome** (Galen)	Non-PEGylated liposome	Daunorubicin	HIV-related Kaposi sarcoma	FDA, 1996
**DepoCyt** (Pacira)	Non-PEGylated liposome	Cytarabine	AML, non-Hodgkin lymphoma	FDA,1999
**Myocet** (Teva UK)	Non-PEGylated liposome	Doxorubicin	Metastatic breast cancer	EMA, 2000
**Marqibo** (Spectrum)	Non-PEGylated liposome	Vincristine	Ph-ALL, Non-Hodgkin’s lymphoma	FDA, 2012
**Onivyde** (Merrimack)	PEGylated liposome	irinotecan	Breast, pancreatic, sarcomas, or brain	FDA, 2015
*Polymer-based*				
**Oncaspar** (Sigma Tau)	PEGylation	L-asparaginase	ALL	FDA,1994
**Abraxane** (Celgene)	Albumin-bound polymer	Paclitaxel	Metastatic pancreatic cancer	FDA, 2005

**Table 5 nanomaterials-11-01727-t005:** Summary of current clinical trials of nanomedicine.

	Name	Nanocarrier	Drug	Clinical Application	Phase	Clinical Trial Number
**Passive**	**NK 105**	Micellar nanoparticle	Paclitaxel	Metastatic or Recurrent Breast Cancer	Phase III	NCT01644890
	**EndoTAG-1**	Cationic liposomes	Paclitaxel	HER2-negative Breast Cancer	Phase II	NCT01537536
				Liver Cancer and Neoplasm Metastasis	Phase II	NCT00542048
	**Nab-rapamycin (ABI-009)**	Albumin-bound nanoparticles	Rapamycin	Solid Tumors	Phase I	NCT00635284
				Non-Muscle Invasive Bladder Cancer (NMIBC)	Phase I/II	NCT02009332
				Malignant Perivascular Epithelioid Cell Tumors	Phase II	NCT02494570
	**CRLX-101 (IT-101)**	Cyclodextrin-based polymer	Camptothecin	Solid Tumor, and Ovarian Cancer	Phase II	NCT00753740
				Non-Small Cell Lung Cancer	Phase II	NCT01380769
	**Nano-luteolin**	PEGylated polymer	Luteolin	Tongue Neoplasms, and Carcinoma	Phase I	NCT03288298
	**NC-6300**	PEGylated polymer	Epirubicin	Solid Tumor and Metastatic Sarcoma	Phase I/II	NCT03168061
	**IT-141**	PEGylated polymer	SN-38 (metabolite of irinotecan)	Cancer, and Recurrent Solid Tumors	Phase I	NCT03096340
**Active**	**BIND-014**	PSMA-targeting polymer (prostate-specific membrane antigen)	Docetaxel	Prostate Cancer	Phase II	NCT01812746
				Non-Small Cell Lung Cancer	Phase II	NCT01792479
				Metastatic Cancer and Solid Tumor	Phase I	NCT01300533
	**MBP-426**	TfR-targeting liposome	Oxaliplatin	Cancer	Phase I	NCT00355888
	**Anti-EGFR-IL-dox**	EGFR-targeting liposome	Doxorubicin	Solid Tumors	Phase I	NCT01702129
				Breast Cancer	Phase II	NCT02833766
	**ThermoDox**	Therapeutic directed (thermally sensitive liposome)	Doxorubicin	Breast Cancer	Phase I/II	NCT00826085
				Colon Cancer Liver Metastasis	Phase II	NCT01464593
				Hepatocellular Carcinoma	Phase III	NCT02112656

## Data Availability

Not applicable.

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
