# Peer review of "From Nanoparticles to Cancer Nanomedicine: Old Problems with New Solutions"

_nanomaterials, 2021, doi:10.3390/nano11071727_

Round 1
Reviewer 1 Report
The authors of the manuscript entitled ''From Nanoparticles to Cancer Nanomedicine: Old problems with new solutions'' have tried to review the current literature on a rapid evolving field of nanomaterials used in the clinic.
Major comments:
1. The authors in general, have tried to incorporate in this review article many different aspects of the nanomaterials used in the clinic, like the liposomes, the exosomes, the carbon nanostructures etc. This tremendous effort of going through the vast and constantly expanding literature, is skewed however, by major syntactic errors and sentences that were too complicated to follow. I would therefore suggest to edit the text with a native english speaker.
2. One major topic that is omitted and should be also included in this review article is the topic of ''theranostic nanomaterials''.
3. Last but not least, nanomedicine has also advanced in the novel field of the immunotherapy of cancer. It would be a small, but a very significant addition to this review, a paragraph related to this topic.
Minor comments:
- Try to summarize on a table the nanomaterials based on their different properties ( liposomes or exosomes or nanohorns etc ) with the different drug targets and the associated cancer type.
Author Response
Reviewer1:
Major comments:
- The authors in general, have tried to incorporate in this review article many different aspects of the nanomaterials used in the clinic, like the liposomes, the exosomes, the carbon nanostructures etc. This tremendous effort of going through the vast and constantly expanding literature, is skewed however, by major syntactic errors and sentences that were too complicated to follow. I would therefore suggest to edit the text with a native English speaker.
We appreciate the comment from Reviewer 1, we had our manuscript edited by a native English speaker as the new version attached.
- One major topic that is omitted and should be also included in this review article is the topic of ''theranostic nanomaterials.''
We are grateful for the suggestion of Reviewer 1 about the important direction of theranostic nanoparticle. However, currently it still has a big gap between theranostic nanoparticle and nanomedicine. The main challenge is the very different aspect of therapy and diagnosis as using the nanoparticles. Generally, the therapeutic nanoparticles are used for relatively longer treatment window in human body which requires low tolerance of nano-bio interaction, while the diagnostic nanomaterials could be one-dose and real-time imaging every half year. We added the discussion of theranostic nanomaterial in the Conclusion section in Page 13, Line 547-567.
- Last but not least, nanomedicine has also advanced in the novel field of the immunotherapy of cancer. It would be a small, but a very significant addition to this review, a paragraph related to this topic.
We appreciate the comment from Reviewer 1. We added the immunogenicity for each material in the main section (Page 9, Line 386-392, and Page 10, Line435-440) and an overview in the Summary section (Page 12, Line 491-500).
Minor comments:
- Try to summarize on a table the nanomaterials based on their different properties (liposomes or exosomes or nanohorns etc ) with the different drug targets and the associated cancer type.
We agree the suggestion of Reviewer 1. We summarized the different properties of each nanomaterials in the Table. 3 and included them in the Page 5, Line164-172. And the treatment targets and therapeutic agents or drug versus associate cancer types are summarized in Table. 4 in Page 6, Line 216-237.
Reviewer 2 Report
This is a very good review. In my opinion, the only drawback of this work is that it is a little bit wordy. The concept of this survey is very good. It is up to day with a huge number of research works. Well done.
Author Response
Reviewer 2:
This is a very good review. In my opinion, the only drawback of this work is that it is a little bit wordy. The concept of this survey is very good. It is up today with a huge number of research works. Well done.
We appreciate the comment from reviewer 2. We had the manuscript edited by native English speakers to shorten several sentences. We hope this review can help our audience catch up the newly update of nanomedicine.
Reviewer 3 Report
This manuscript addresses the status of development of approved nanomedicines and opportunities for novel materials that combine new findings in tumor and nanotechnology to develop more effective nanomedicines for cancer patients. The paper is well written and has the merit of publication. However, the authors could provide future perspectives and a brief overview of next steps in the field.
Some typos/bold could be reviewed.
Author Response
Reviewer3:
This manuscript addresses the status of development of approved nanomedicines and opportunities for novel materials that combine new findings in tumor and nanotechnology to develop more effective nanomedicines for cancer patients. The paper is well written and has the merit of publication. However, the authors could provide future perspectives and a brief overview of next steps in the field.
Some typos/bold could be reviewed.
We appreciate the comment of reviewer 3. To have our audience better realize the nanomedicine field, we newly added the perspective overview in the Conclusion section in Page 13, Line 547-567.
Round 2
Reviewer 1 Report
The authors have made significant improvements to the written language. Although, there are still a lot of phrases that must be corrected. I would suggest to make one last effort to correct the remaining errors and typos.
INDICATIVE MAJOR CORRECTIONS:
Please consider rephrasing or removing this first sentence of the introduction,''Nanoparticles are widely used in physiological delivery with its strong penetration''.
What is the meaning of the phrase in line 393 : The potential of DNA origami constructs was tested, showing programmable, non-cytotoxicity, but recognition by Toll-like receptor 9 (TLR9) which is a receptor for innate immune system localized in the endosome? Please rephrase.
LINE 400 : ''Although a great strength under physiological pH condition,''
The verb is missing.
LINE 403: To prevent degradation and stabilize the structure, couple modification can be applied to the nucleotide scaffold. (couple modification (s) could be applied.
LINE 414: Zhao’s group has a very exciting design of intravenously injected DNA nanorobots. (has developed)
LINE 425 :The mechanism was further suggested the exosome is a major route of excretion, which allows cells to exchange biomaterials, such as RNA, proteins and lipids. .. Rephrase.
There are still many phrases to be corrected throughout the manuscript...Please revise them thoroughly.
Author Response
Response to general comments:
We thank the reviewer for the helpful suggestions and we have addressed all of the comments, and highlighted the changes in the manuscript. In addition, the language has been revised and the text polished up by a native English speaker.
INDICATIVE MAJOR CORRECTIONS:
- Please consider rephrasing or removing this first sentence of the introduction, ''Nanoparticles are widely used in physiological delivery with its strong penetration''.
Response: The sentence has been removed in the manuscript.
- What is the meaning of the phrase in line 393: The potential of DNA origami constructs was tested, showing programmable, non-cytotoxicity, but recognition by Toll-like receptor 9 (TLR9) which is a receptor for innate immune system localized in the endosome? Please rephrase.
Response:
- LINE 400 : ''Although a great strength under physiological pH condition,''The verb is missing.
Response: We have revised the sentences into “Deposit a great strength under physiological pH condition, the structure of DNA ori-gami has relative limited thermal and chemical stability.”
- LINE 403: To prevent degradation and stabilize the structure, couple modification can be applied to the nucleotide scaffold. (couple modification (s) could be applied.
Response: We have revised the sentences into “To prevent degradation and stabilize the structure, a couple modifications could be applied to the nucleotide scaffold.”
- LINE 414: Zhao’s group has a very exciting design of intravenously injected DNA nanorobots. (has developed)
Response: We have revised the sentences into Zhao’s group has developed a very exciting design of intravenously injected DNA nanorobots.
- LINE 425: The mechanism was further suggested the exosome is a major route of excretion, which allows cells to exchange biomaterials, such as RNA, proteins and lipids.
Response: We have revised the sentences as “The main function of exosome was suggested as a route of cellular communication, which allows cells to exchange biomaterials, such as RNA, proteins, and lipid components.”
- There are still many phrases to be corrected throughout the manuscript...Please revise them thoroughly.
Response: We thank the reviewer for the suggestion, and have corrected the phrases throughout the manuscript.